# Distinct memory engrams in the infralimbic cortex of rats control opposing environmental actions on a learned behavior

Nobuyoshi Suto[1]*[†], Amanda Laque[1†], Genna L De Ness[1], Grant E Wagner[1], Debbie Watry[1], Tony Kerr[1], Eisuke Koya[2], Mark R Mayford[3], Bruce T Hope[4]*, Friedbert Weiss[1]*

[1]Department of Molecular and Cellular Neuroscience, The Scripps Research Institute, La Jolla, United States; [2]Sussex Neuroscience, School of Psychology, University of Sussex, Brighton, United Kingdom; [3]Department of Psychiatry, University of California San Diego, San Diego, United States; [4]Behavioral Neuroscience Branch, National Institute on Drug Abuse/National Institutes of Health/Intramural Research program, Baltimore, United States

**Abstract** Conflicting evidence exists regarding the role of infralimbic cortex (IL) in the environmental control of appetitive behavior. Inhibition of IL, irrespective of its intrinsic neural activity, attenuates not only the ability of environmental cues predictive of reward availability to promote reward seeking, but also the ability of environmental cues predictive of reward omission to suppress this behavior. Here we report that such bidirectional behavioral modulation in rats is mediated by functionally distinct units of neurons (neural ensembles) that are concurrently localized within the same IL brain area but selectively reactive to different environmental cues. Ensemble-specific neural activity is thought to function as a memory engram representing a learned association between environment and behavior. Our findings establish the causal evidence for the concurrent existence of two distinct engrams within a single brain site, each mediating opposing environmental actions on a learned behavior.

**\*For correspondence:** nsuto@ scripps.edu (NS); bhope@intra. nida.nih.gov (BTH); bweiss@ scripps.edu (FW)

[†]These authors contributed equally to this work

**Competing interests:** The authors declare that no competing interests exist.

## Introduction

Different units of neurons or 'neural ensembles' (*Eichenbaum, 1993*) within a single brain area selectively react to different environmental cues, as shown by electrophysiological and calcium imaging studies (e.g., *Schoenbaum et al., 1998*; *Jog et al., 1999*; *Frank et al., 2000*; *Ohki et al., 2005*; *Komiyama et al., 2010*). Such ensemble-specific brain activity is thought to function as the memory engram, each representing a unique learned association between environment and behavior (*Cruz et al., 2013*; *Mayford, 2014*; *Tonegawa et al., 2015*). Based on the correlational evidence from electrophysiological and calcium imaging studies, it is commonly thought that multiple engrams concurrently exist within a single brain site at a single stage of learning. Previous studies have established the causal evidence that a single brain area (*Warren et al., 2016*) or ensemble (*Redondo et al., 2014*) can represent two different engrams at two different stages of learning (e.g., before vs. after extinction training). However, the causal evidence supporting the concurrent existence of multiple engrams within a single brain site, each mediating different environmental actions on a learned behavior (e.g., a go/no-go task), is still lacking.

Here we present the causal evidence for the concurrent existence of two distinct engrams within the infralimbic cortex (IL) of rats, each mediating opposing environmental actions on a pre-established operant response. Our findings help reconcile the contradictory reports regarding the role of IL in the environmental control of appetitive behavior (*Gourley and Taylor, 2016*). Some reports (*Koya et al., 2009a*; *Bossert et al., 2011*) indicate that non-selective disruption of IL, irrespective of neural activity, prevents promotion of reward seeking by cues signaling reward availability (e.g., reward contexts). However, other reports (*Peters et al., 2008*; *LaLumiere et al., 2012*) indicate that non-selective disruption of IL prevents suppression of reward seeking by cues signaling reward omission (e.g., extinction contexts). Yet other reports (*Willcocks and McNally, 2013*; *Pfarr et al., 2015*) indicate that the non-selective disruption of IL has no significant effect on either promotion or suppression of reward seeking.

These discrepancies could be the results of different IL neurons selectively reacting to different cues signaling reward availability and omission. We hypothesized that distinct neural ensembles – concurrently existing within IL but each selectively reactive to a discriminative stimulus predictive of reward availability (S+) or omission (S-) – differentially control promotion and suppression of reward seeking. To test this hypothesis, we developed an animal model for bidirectional environmental modulation of reward seeking, and utilized a neural activity-targeted disruption technique (Daun02 pharmacogenetic inactivation in *Fos-lacZ* transgenic rats) (*Koya et al., 2009b*) to inhibit neural ensembles in IL specifically activated by either S+ or S- cues.

## Results and discussion

All rats were trained to recognize three sets of cues predictive of rewards at varying probability: active lever and light-cue (50%), S+ (100%) and S- (0%). All rats were trained to lever-press for a sweet solution (containing 3% glucose and 0.125% saccharin) presented together with a light-cue (*Figure 1A*), and further trained to recognize distinct auditory cues (white noise and beeping tone)

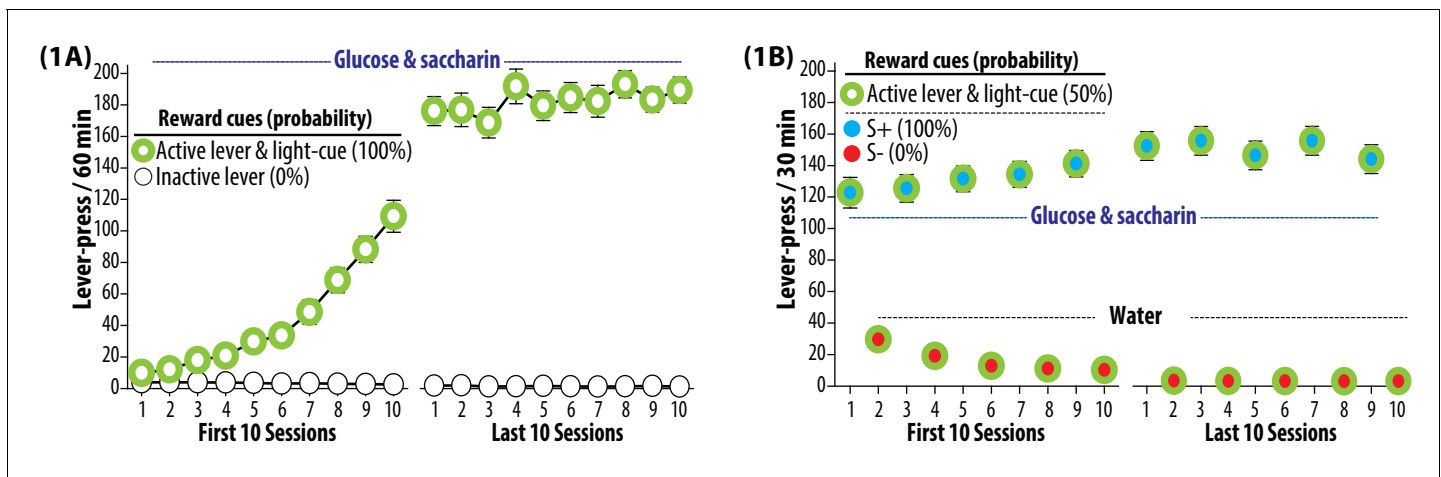

**Figure 1.** Trainings. (**1A**) Self-administration training to establish operant response for a palatable reward. The rats were subjected to once daily operant conditioning sessions (60 min, each) to press an 'active lever' for the purpose of gaining a sweet solution (3% glucose and 0.125% saccharin dissolved in water). Each delivery of the sweet solution was paired with a 'light-cue'. Presses on an 'inactive lever' were without a scheduled consequence. Each rat was required to satisfy preset training criteria, and underwent a total of 35 to 42 sessions. Group means of the total number of lever presses per session (±SEM) during the first and last 10 sessions are depicted. During this phase, both active lever and light-cue predicted the availability of sweet solution 100% of the time. N = 90. (**1B**) Discrimination training to establish two distinct auditory cues (white noise and beeping tone) as the discriminative stimuli predictive of reward availability (S+) and omission (S-). Each rat was subjected to alternating once daily operant conditioning sessions (30 min, each) to press the active lever for the purpose of gaining the sweet solution (preceded and accompanied by the S+ auditory cue) or plain water (preceded and accompanied by the S- auditory cue). Each delivery of either sweet solution or water was paired with the light-cue. During this phase, each experimentally manipulated stimulus was conditioned to predict glucose and saccharin at the following probabilities: active lever and light-cue (50%), S+ (100%) and S- (0%). Each rat was required to satisfy three preset training criteria, and underwent a total of 92 to 98 sessions (46 to 49 S+ sessions and 46 to 49 s- sessions). Group means of the total numbers of lever presses (±SEM) recorded during the first and last 10 sessions are depicted. N = 90.

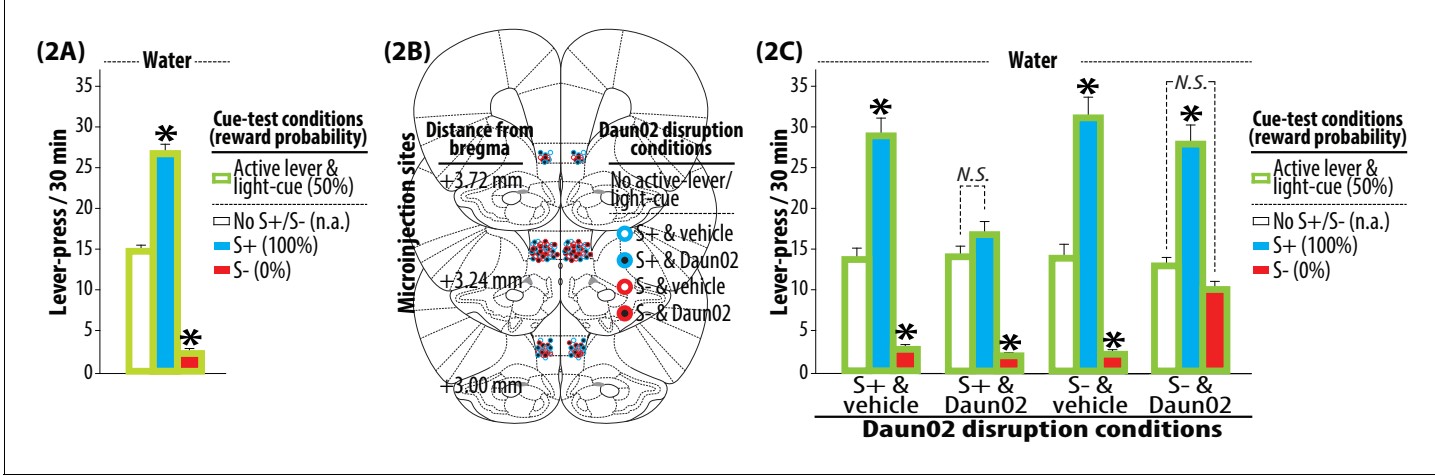

**Figure 2.** Tests and Daun02 disruption. (**2A**) Pre-disruption tests for the bidirectional modulation of reward seeking by S+ or S-. Each rat was subjected to three once daily cue-test conditions (30 min, each): (1) active lever and light-cue without S+ or S- (No S+/S-), (2) active lever, light-cue and S+, (3) active lever, light-cue and S-. Under all cue-test conditions, a press on the active lever delivered water and illuminated the light-cue. The sweet solution was not available. Group means of the total number of active lever-presses under each cue-test condition (+SEM) are depicted. N = 90. *p<0.001 (vs. No S+/S-). (**2B**) Daun02 disruption for the activity-targeted inhibition of neural ensembles in IL reactive to S+ or S-. The rats were randomly assigned to one of four experimental groups defined by the type of disruption cue (S+ or S-) and microinjection (Daun02 or vehicle). The rats were exposed to S+ or S- (90 min), and then received a bilateral microinjection of Daun02 (2.0 μg/0.5 μl/side) or vehicle (0.5 μl/side) into IL. In order to disrupt neurons specifically activated by S+ or S-, both active lever and light-cue were withheld ('No active lever/light-cue'). Representative sites of microinjection are depicted. (**2C**) Post-disruption tests for the bilateral modulation of reward seeking by S+ or S-. Each rat was subjected to three once daily cue-test conditions. The experimental schedule effective during the Pre-disruption tests (**2A**) was applied. Group means of the total number of active lever-presses under each cue-test condition (+SEM) are depicted. N = 21–24, each. *p<0.01–0.001 (vs. No S+/S-).

as S+ and S- (*Figure 1B*). Each rat underwent three cue-tests (*Figure 2A*) for bidirectional environmental modulation of reward seeking: (1) active lever and light-cue without S+ or S- ('No S+/S-'), (2) active lever, light-cue and S+, (3) active lever, light-cue and S-. Glucose and saccharin were not available during these tests. One-way analysis of variance (ANOVA) on active lever-pressing revealed a significant effect of cue-test ($F_{(2,89)}$ = 216.679, p<0.001). The active lever and light-cue sufficiently initiated and maintained lever-pressing (see No S+/S-). In each animal, superimposing S+ significantly potentiated lever-pressing (p<0.001), while superimposing S- significantly suppressed this behavior (p<0.001). This bidirectional environmental modulation of reward seeking is presumably mediated by learning processes known as 'conditioned excitation or inhibition' (*Rescorla, 1969*) or 'positive or negative occasion setting' (*Schmajuk et al., 1998*).

The rats were then randomly divided into four groups defined by disruption-cue (S+ or S-) and microinjection (Daun02 or vehicle) for neural activity-targeted inactivation (*Figure 2B*) using previously established procedures (*Koya et al., 2009b*; *Bossert et al., 2011*; *Pfarr et al., 2015*). Each rat was first exposed to either S+ or S-, and then received a bilateral microinjection of Daun02 (2.0 μg/0.5 μl/side) or vehicle (0.5 μl/side) into IL. In *Fos-lacZ* rats, Daun02 (inactive compound) is catalyzed into daunorubicin (cytotoxin) by beta-galactosidase (enzyme) only in 'activated' cells expressing Fos (activation marker), thereby triggering apoptosis (see *Pfarr et al., 2015*). In contrast, Daun02 cannot be catalyzed into daunorubicin in 'non-activated' cells lacking Fos expression (and thus beta-galactosidase), and cellular disruption is prevented. As such, Daun02 selectively inactivates Fos-positive 'activated' neurons without disrupting the surrounding Fos-negative 'non-activated' neurons. On the day of Daun02 disruption, both active lever and light-cue were not introduced. Thus, the rats were withheld from lever-pressing, as to not disrupt neurons activated by either active lever or light-cue.

Each rat then again underwent the three cue-tests for the bidirectional modulation of reward seeking by S+ or S- (*Figure 2C*). Three-way ANOVA on active lever-press revealed significant effects of disruption-cue ($F_{(1,86)}$ = 8.839, p<0.01), cue-test ($F_{(2,172)}$ = 313.899, p<0.001), and significant interactions for disruption-cue and microinjection ($F_{(1,86)}$ = 5.87, p<0.05), disruption-cue and cue-test ($F_{(2,172)}$ = 9.694, p<0.001), microinjection and cue-test ($F_{(2,172)}$ = 22.936, p<0.001), and disruption-

cue and microinjection and cue-test ($F_{(2,172)}$ = 6.172, p<0.01). In the two vehicle-treated control groups ('S+ & vehicle' and 'S- & vehicle'), the bidirectional modulation of lever-pressing by S+ or S- was evident (p<0.01–0.001). In the group that received Daun02 following exposure to S+ ('S+ & Daun02'), Daun02 disruption of IL neurons activated by S+ blocked the promotion of lever-pressing by S+ but spared the suppression of this behavior by S- (p<0.01). In the group that received Daun02 following exposure to S- ('S- & Daun02'), Daun02 disruption of IL neurons activated by S- blocked suppression by S- but spared promotion by S+ (p<0.01). For all cases, basal lever-pressing initiated and maintained by active lever and light-cue was not significantly altered by Daun02 disruption of IL neurons activated by S+ or S- (see No S+/S-). Thus, Daun02 disruption of IL neurons activated by either S+ or S- exclusively altered the cue-modulation of reward seeking uniquely linked to the targeted cue in a single animal.

Daun02's effect on disrupting neural ensemble activity was verified by Fos immunohistochemistry (*Figure 3*) using previously established procedures (*Koya et al., 2009b*; *Bossert et al., 2011*; *Pfarr et al., 2015*). Each of the four groups prepared for the Daun02 disruption was further divided into three groups defined by the final Fos-induction conditions ('S+', 'S-' or 'No S+/S-'). Each rat was subjected to one of these conditions. However, active lever and light-cue were not introduced in order to determine IL ensembles specifically reactive to S+ or S-. Three-way ANOVA on Fos-positive nuclei in IL revealed significant effects of microinjection ($F_{(1,78)}$ = 7.695, p<0.01), Fos-induction ($F_{(2,78)}$ = 31.129, p<0.001), and significant interactions for disruption-cue and Fos-induction ($F_{(2,78)}$ = 14.803, p<0.001), disruption-cue and microinjection and Fos-induction ($F_{(2,78)}$ = 11.016, p<0.001). In the two vehicle-treated control groups ('S+ & vehicle' and 'S- & vehicle'), both S+ and S- significantly increased Fos-positive nuclei in IL (Ps < 0.05–0.01). In the group that received Daun02 following exposure to S+ ('S+ & Daun02'), Daun02 disruption of IL neurons activated by S+ prevented the induction of Fos by S+ but spared the induction by S- (p<0.05). In the group that received Daun02 following exposure to S- ('S- & Daun02'), Daun02 disruption of IL neurons activated by S- significantly reduced the induction of Fos by S- but spared the induction by S+ (p<0.05). Thus, Daun02 disruption of IL neurons activated by either S+ or S- exclusively altered the cue-triggered neural activation uniquely linked to the targeted cue in a single animal.

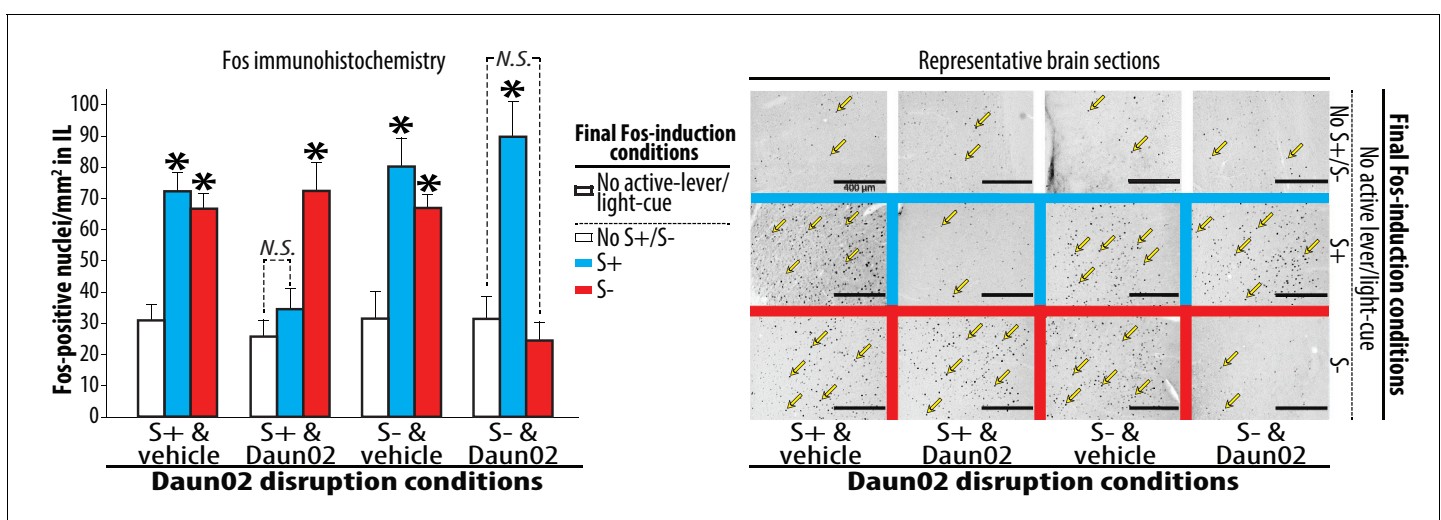

**Figure 3.** Neural ensembles in IL reactive to S+ or S- following the activity-targeted disruption by Daun02. Each of the four experimental groups, prepared and used for the Daun02 disruption (*Figure 2B*) and Post-disruption tests I (*Figure 2C*), was further randomly divided into three groups defined by the type of final Fos induction condition (S+, S- or No S+/S-). The rats were exposed to S+, S- or No S+/S- (control), then deeply anesthetized and euthanized. Brains were collected, sectioned (40 µm), processed for Fos immunohistochemistry. Fos-positive nuclei from sampling areas around the IL microinjection sites were quantified double-blindly. The average numbers of Fos-positive nuclei per mm² were calculated for each rat, and used for statistical analyses. Yellow arrows represent typical Fos-positive nuclei. Group means of these average numbers (+SEM) are depicted. N = 7–9, each. *p<0.05–0.01 (vs. No active lever/light-cue).

Together with a recent electrophysiological report (*Moorman and Aston-Jones, 2015*), our findings establish that the same IL brain area is capable of controlling both promotion and suppression of reward seeking via different neural ensembles, each selectively reactive to S+ or S-. Because Daun02 disruption of IL neurons activated by either S+ or S- exclusively altered the behavioral response (*Figure 2*) as well as neural activation (*Figure 3*) uniquely linked to the targeted cue, mutually exclusive rather than overlapping neural ensembles in IL likely mediate the bidirectional environmental control of reward seeking. As is the case of recent studies that utilized similar activity-dependent procedures (e.g., *Redondo et al., 2014*; *Grosso et al., 2015*), the current results demonstrate the functional significance of manipulating brain cells based on their activity in addition to other characteristics, such as locality and phenotype. We thus raise caution to the use of non-selective techniques that manipulate neural activity irrespective of intrinsic neural activity for determining the brain behavioral functions. Finally, our findings provide the causal evidence for the concurrent existence of multiple engrams, each mediating different and even opposing environmental actions on a behavior, in a single brain area. In light of recent studies implicating differential brain processes in differential environmental control of a learned response (*Ciocchi et al., 2010*; *Trouche et al., 2013*; *Adhikari et al., 2015*), ensemble-specific neural phenotype as well as both local and brain-wide neurocircuitry involving each unique engram should be elucidated in future studies.

## Materials and methods

### Subjects

A total of one hundred nineteen male *Fos-lacZ* transgenic rats (*Kasof et al., 1995*) on a Sprague-Dawley background were used (RRID:SCR_014785). Of which, a total of ninety rats were retained for statistical analyses (see below). This transgenic strain is required for neural activity-targeted disruption of neural ensembles by Daun02 pharmacogenetic inactivation (*Koya et al., 2009b*). All rats were bred at The Scripps Research Institute, and genotyped by Laragen, Inc. (Culver City, CA). The rats were group-housed (two rats per cage) in plastic cages in a temperature- and humidity-controlled room, maintained on a 12 hr/12 hr reverse light/dark cycle (lights on at 20:00 hr). At all times, the rats were allowed free access to food and water.

### Surgery

All *Fos-lacZ* rats (250–300 g) were surgically implanted with permanent bilateral guide cannulae (22 G; Plastics One, Roanoke, VA, USA) under isoflurane anesthesia for the microinjection of Daun02 or vehicle (see below) into the infralimibic cortex (IL). The microinjection coordinates were anteroposterior + 3.2 mm, mediolateral ± 0.6 mm, dorsoventral −5.5 mm. Rats were allowed to recover at least seven days before the start of the behavioral procedures (see below). One rat died during the surgery, and five rats died due to post-operation complications.

### Behavioral procedures

We developed an animal model for the bidirectional modulation of reward seeking by discriminative stimuli predictive of reward availability (S+) and reward omission (S-). General schematics and timeline are depicted in *Figure 4*. This model consisted of twelve experimental phases and lasted over six months. The rats were always trained and tested during the dark (active) phase in a dedicated operant conditioning chamber ('chamber') equipped with two retractable levers (one 'active lever' and one 'inactive lever'), a 'light-cue' and a drinking well. At all times, insertion of both active and inactive levers signaled the start of a once daily operant conditioning session conducted under a fixed ratio 1 schedule of reinforcement. During this session, a press on the active lever resulted in a single delivery of 0.2 ml of either sweet solution containing 3% glucose and 0.125% saccharin dissolved in water ('glucose & saccharin') or plain water ('water') into the drinking well. In our pilot experiments (data not included), this combination of glucose and saccharin produced more robust operant responding than solutions containing either glucose or saccharin alone. Each delivery of either sweet solution or water was paired with 5s illumination of the light-cue signaling a 5s time-out period. During the 5s time-out period, presses on the active lever were recorded but without a scheduled consequence. At all times, presses on the inactive lever were recorded but without a scheduled consequence. Throughout the course of this model, the rats were repeatedly subjected

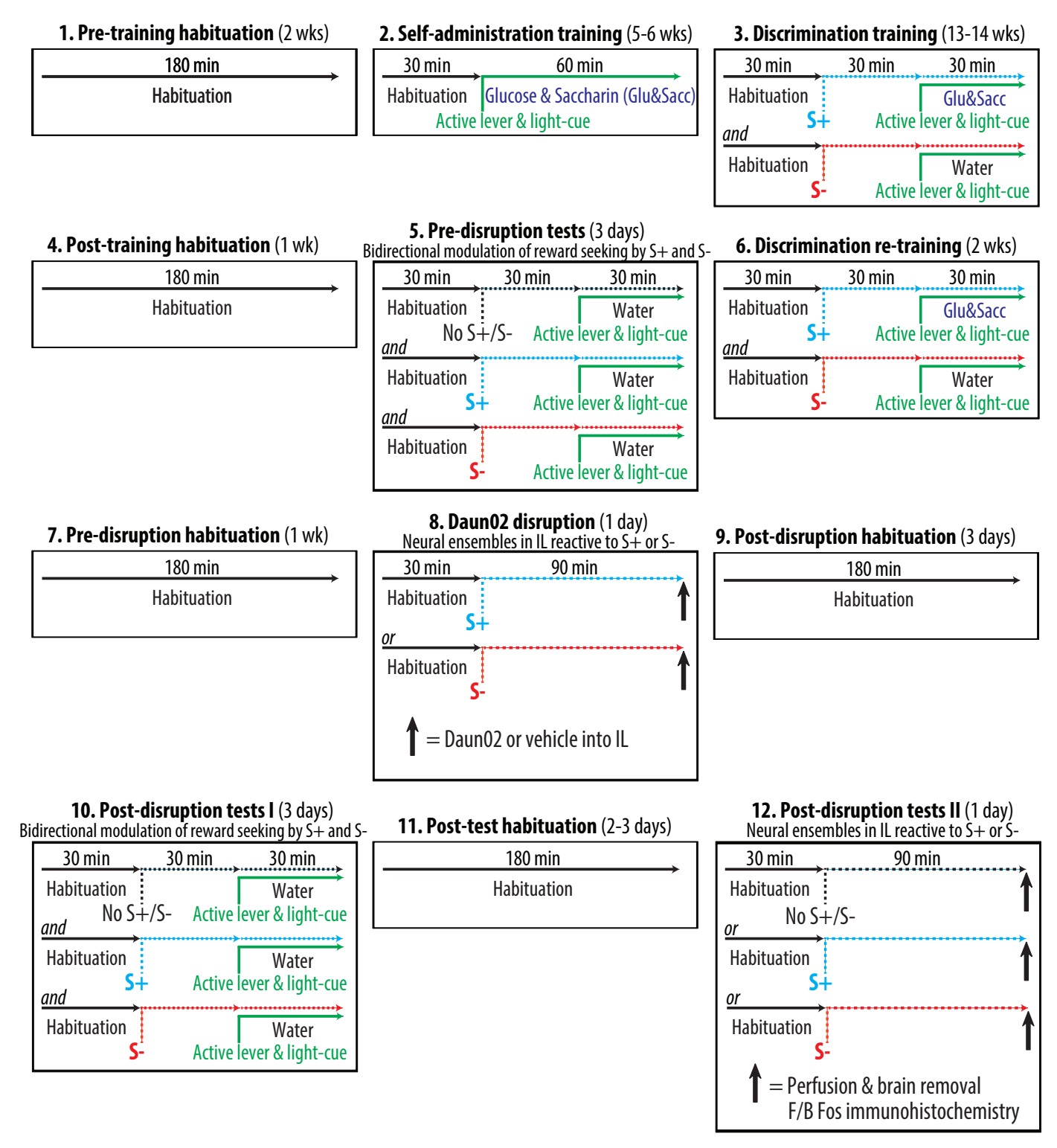

**Figure 4.** Experimental time course and schedule.

to habituation periods (both between-sessions and within-sessions) in order to minimize the neuro-behavioral impact of environmental cues (e.g., experimenters and chambers) other than those

manipulated experimentally (i.e., active lever, light-cue, S+ and S-). Three different cohorts of rats were subjected to the entire behavioral procedure (i.e., three biological replicates).

Detailed procedures for each experimental phase are described below:

### Pre-training habituation (two weeks)

The aim of this phase was to minimize the neurobehavioral impact of environmental cues other than those manipulated experimentally. All rats were placed in the chamber daily for 180 min without a scheduled consequence.

### Self-administration training (5–6 weeks)

The aim of this phase was to establish appetitive operant responses in each animal. All rats were trained daily to lever press for the sweet solution in once daily sessions. The rats were placed in the chamber and habituated for 30 min. Insertion of both active and inactive levers then marked the start of a 60 min session to self-administer glucose and saccharin. A press on the active lever delivered the sweet solution, and also illuminated the light-cue. Each rat was required to satisfy two pre-set training criteria: a minimum of five weeks of training (35 sessions), and a minimum of 100 deliveries of the sweet solution for three consecutive sessions. Three rats that did not satisfy these criteria within six weeks of training were excluded. During this phase, the insertion of the active lever and the illumination of the light-cue predicted the availability of the sweet solution 100% of the time.

### Discrimination training (13–14 weeks)

The aim of this phase was to establish discriminative stimuli predictive of reward availability (S+) and reward omission (S-) in each animal. All rats were trained daily to recognize two types of auditory cues (white noise and beeping tone) as S+ and S-. The assignment of white noise and beeping tone as S+ or S- was counterbalanced between subjects. Each rat was subjected to alternating once daily sessions to self-administer the sweet solution (preceded and accompanied by the S+ auditory cue) or water (preceded and accompanied by the S- auditory cue). The rats were placed in the chamber and habituated for 30 min. One of the auditory cues was then introduced starting 30 min prior to the insertion of both levers, and maintained throughout a 30 min session to lever press for either the sweet solution or water. Each rat was required to satisfy three preset training criteria: a minimum of 13 weeks of training (i.e., 46 S+ sessions and 46 S- sessions), a minimum of 50 deliveries of the sweet solution for three consecutive S+ sessions, and a maximum of five deliveries of water for three consecutive S- sessions. Nine rats that did not satisfy these criteria within 14 weeks of training were excluded. During this phase, each experimentally manipulated stimulus was conditioned to predict the sweet solution at varying probability: active lever and light-cue (50%), S+ (100%) and S- (0%).

### Post-training habituation (one week)

The aim of this phase was to minimize the neurobehavioral impact of environmental cues other than those manipulated experimentally. All rats were placed in the chamber daily for 180 min without a scheduled consequence.

### Pre-disruption tests (three days)

The aim of this phase was to establish the baseline measurements of the bidirectional modulation of reward seeking by S+ and S- in each animal. All rats were tested daily for the bilateral environmental modulation of reward seeking. Each rat underwent three once daily cue-tests: [1] active lever and light-cue without S+ or S- (No S+/S- test), [2] active lever, light-cue and S+ (S+ test), [3] active lever, light-cue and S- (S- test). The order of these conditions was randomly counterbalanced between subjects. For all cue-tests, a press on the active lever delivered water and illuminated the light-cue. The sweet solution was not available. For the No S+/S- test, each rat was placed in the chamber and habituated for 60 min. Insertion of both active and inactive levers then marked the start of a 30 min cue-test. Neither S+ nor S- was introduced. For the S+ test, each rat was placed in the chamber and habituated for 30 min. The S+ auditory cue was then introduced starting 30 min prior to the insertion of levers, and maintained throughout a 30 min cue-test. For the S- test, each rat was placed in the chamber and habituated for 30 min. The S- auditory cue was then introduced starting 30 min prior to the insertion of levers, and maintained throughout a 30 min cue-test.

### Discrimination re-training (two weeks)

The aim of this phase was to establish discriminative stimuli predictive of reward availability (S+) and reward omission (S-) in each animal. All rats were re-trained daily for S+ and S- following the experimental schedules effective during the discrimination training.

### Pre-disruption habituation (one week)

The aim of this phase was to minimize the neurobehavioral impact of environmental cues other than those manipulated experimentally. All rats were placed in the chamber daily for 180 min without a scheduled consequence.

### Daun02 disruption (one day)

The aim of this phase was to permanently disrupt IL neurons selectively reactive to S+ or S- in different animals. Neural ensembles in IL reactive to S+ or S- were permanently disrupted (lesion by apoptosis) by Daun02 pharmacogenetic inactivation technique (*Koya et al., 2009b*; *Pfarr et al., 2015*). The rats were randomly assigned to one of four experimental groups defined by disruption-cue (two levels: S+ or S-) and microinjection (two levels: Daun02 or vehicle). All rats were placed in the chamber and habituated for 30 min. The disruption cue (S+ or S-) was then introduced. The rats remained in the chamber for additional 90 min in the presence of S+ or S-. This timing (90 min) was to optimally induce β-galactosidase and Fos in neurons activated by either S+ or S- (*Koya et al., 2009b*). In order to disrupt the neurons specifically activated by S+ or S-, both levers and light-cue were not introduced, and thus the rats were withheld from lever pressing. The rats then received a bilateral microinjection of Daun02 (2.0 μg/0.5 μl/side) or vehicle (0.5 μl/side) into IL, and returned to their homecages. The Daun02 disruption was based on previously published procedures (e.g., *Koya et al., 2009b*; *Bossert et al., 2011*; *Pfarr et al., 2015*). We purchased Daun02 (*Farquhar et al., 2002*) from Sequoia Research Products (Pangbourne, Berkshire, UK: Cat# SRP0400g).

### Post-disruption habituation (three days)

The aim of this phase was to minimize the neurobehavioral impact of environmental cues other than those manipulated experimentally. All rats were placed in the chamber daily for 180 min without a scheduled consequence.

### Post-disruption tests I (three days)

The aim of this phase was to determine the behavioral impacts of Daun02 disruption of IL neurons reactive to S+ or S- on the bidirectional modulation of reward seeking by S+ and S-. The effects of Daun02 disruption of IL ensembles on the bilateral modulation of reward seeking by S+ and S- were determined. Each rat was again subjected to the once daily three cue-tests following the experimental schedules effective during the pre-disruption tests.

### Post-behavioral test habituation (2–3 days)

The aim of this phase was to minimize the neurobehavioral impact of environmental cues other than those manipulated experimentally. All rats were placed in the chamber daily for 180 min without a scheduled consequence.

### Post-disruption tests II (one day)

The aim of this phase was to confirm Daun02 disruption of IL neurons specifically reactive to S+ or S-. The rats were prepared and euthanized for the histological verification of Daun02 disruption of IL ensembles. Each of the four experimental groups – prepared for Daun02 disruption and post-disruption tests I – was further randomly divided into three groups (N = 7–9, each, retained for statistical analyses) defined by the type of final Fos-induction condition (three levels: 'S+', 'S-' or 'No S+/S-'). All rats were first placed in the chamber and habituated for 30 min. The rats remained in the chamber for additional 90 min in the presence of the S+ auditory cue, the S- auditory cue or no additional stimulus (No S+/S-). In order to determine the IL ensembles specifically reactive to S+ or S-, both levers and light-cue were not introduced, and the rats were thus withheld from lever pressing. The rats were then

deeply anesthetized with isoflurane and intracardially perfused with 100 ml of 1× PBS followed by 200 ml of fixative solution (phosphate buffer, containing 4% paraformaldehyde and 14% saturated picric acid). Brains were collected, postfixed for 24 hr at 4°C in fixative solution and sectioned (40 μm).

## Fos and β-galactosidase immunohistochemistry

The brain sections harvested from all animals were processed for Fos immunohistochemistry and quantified as described previously (e.g., *Koya et al., 2009b*; *Bossert et al., 2011*; *Pfarr et al., 2015*). Fos antibody (1:2000 dilution) from Cell Signaling Technology (Cat# 2250S, Danvers, MA, USA; RRID: AB_2247211) was used. The sections were developed using an ImmPRESS HRP (Peroxidase) Polymer Kit from Vector Laboratories (Cat# MP-7451, Burlingame, CA, USA; RRID:AB_2631198) and diaminobenzide. β-galactosidase antibody (1:1000 dilution) from Santa Cruz Biotechnology (Cat# sc-65670, Dallas, TX, USA; RRID:AB_831022) was used. Additional brain sections were processed for X-gal (5-bromo-4-chloro-3-indolyl β-D-galactopyranoside) histochemistry to validate the presence of β-galactosidase as described previously (e.g., *Koya et al., 2009b*; *Bossert et al., 2011*; *Pfarr et al., 2015*). We purchased X-gal kit (Cat# XGAL-0100;) from Rockland Immunochemicals Inc. (Pottstown, PA).

## Histological procedures

The histological validation of Daun02 disruption was based on previously published procedures (e.g., *Koya et al., 2009b*; *Bossert et al., 2011*; *Pfarr et al., 2015*). Bright-field images of IL were captured and digitized using EVOS microscope (ThermoFisher, Inc., Waltham, MA, USA). These images were used for (1) histological verification of the co-localization of Fos and β-galactosidase (i.e., Fos-lacZ positive), (2) histological verification of the microinjection sites, and (3) histological quantification of Fos-expressing nuclei. Nuclei expressing Fos and/or β-galactosidase were counted using ImageJ (National Institute of Health, Bethesda, MD, USA; RRID:SCR_003070). The threshold level was set to detect moderate to darkly stained nuclei but not lightly stained nuclei. We counted nuclei from sampling areas around the IL injection site from 3–5 coronal sections per rat. Average numbers of Fos-positive nuclei per $mm^2$ calculated for each rat were used for statistical analyses and data representations. Image capture and quantification were conducted by an observer blind to the experimental conditions. One rat was determined to be *Fos–lacZ* negative (no co-localization of Fos and β-galactosidase), and excluded. Seven rats were excluded because their microinjection sites were determined to be outside of IL. Three rats were excluded due to necrosis in tissues surrounding the microinjection sites.

## Data analysis

We did not use a power analysis to predetermine sample sizes. However, our sample sizes are similar to those reported in previous publications. We used non-parametric statistical methods to reduce the effect of the small sample size. The behavioral results from the pre-disruption and post-disruption tests I were analyzed separately. For each test, the total numbers of active lever presses recorded during the 30 min lever-pressing session were used. For the pre-disruption tests, the statistical analyses were conducted using one-way within analysis of variance (ANOVA) with cue-test (three levels: S+, S- and No S+/S-) as within-group factor. For the post-disruption tests I, the statistical analyses were conducted using two-way between and one-way within ANOVA with disruption-cues (two levels: S+ or S-) and microinjection (two levels: Daun02 or vehicle) as between-group factors and cue-test (three levels: S+, S- and No S+/S-) as within-group factor. For the post-disruption tests II, the average numbers of Fos-positive nuclei per $mm^2$ were calculated for each animal and used for statistical analyses. These analyses were conducted using three-way between ANOVA with the type of disruption-cue (two levels: S+ or S-), microinjection (two levels: Daun02 or vehicle) and Fos-induction (three levels: S+, S- or No S+/S-) as between-group factors. For all cases, Tukey test was used for *post-hoc* group comparisons when appropriate. Effects were considered significant when p<0.05. We used IBM SPSS Statistics (IBM, Armonk, NY, USA; RRID:SCR_002865) and Sigma Plot (Systat Software, San Jose, CA, USA; RRID:SCR_003210) for conducting statistical analyses.

## Acknowledgements

This is publication number 29442 from The Scripps Research Institute. We gratefully thank Dr. Jennifer M Bossert (NIDA/NIH/IRP) for her technical support in immunohistochemistry. This work was supported by the Extramural and Intramural funding from National Institute on Drug Abuse as well as National Institute of Alcohol Abuse and Alcoholism, National Institute of Health, USA: R21DA033533 (NS), R01DA037294 (NS), R01AA023183 (NS), R01AA021549 (FW) and ZIADA000467 (BTH). AL was supported by Ruth L Kirschstein Institutional National Research Service Award from National Institute of Alcohol Abuse and Alcoholism, National Institute of Health, USA: T32AA007456. AL was supported by Ruth L. Kirschstein Institutional National Research Service Award from National Institute on Alcohol Abuse and Alcoholism, National Institute of Health, USA: T32AA007456.

## Additional information

### Funding

| Funder | Grant reference number | Author |
|---|---|---|
| National Institute on Drug Abuse | R21DA033533 | Nobuyoshi Suto |
| National Institute on Alcohol Abuse and Alcoholism | R01AA023183 | Nobuyoshi Suto |
| National Institute on Drug Abuse | R01DA037294 | Nobuyoshi Suto |
| National Institute on Drug Abuse | ZIADA000467 | Bruce T Hope |
| National Institute on Alcohol Abuse and Alcoholism | R01AA021549 | Friedbert Weiss |

The funders had no role in study design, data collection and interpretation, or the decision to submit the work for publication.

### Author contributions

NS, Conception and design, Acquisition of data, Analysis and interpretation of data, Drafting or revising the article; AL, Acquisition of data, Analysis and interpretation of data, Drafting or revising the article; GLDN, GEW, DW, TK, Acquisition of data, Drafting or revising the article; EK, Analysis and interpretation of data, Drafting or revising the article; MRM, FW, Conception and design, Analysis and interpretation of data, Drafting or revising the article; BTH, Conception and design, Analysis and interpretation of data, Drafting or revising the article, Contributed unpublished essential data or reagents

### Author ORCIDs

Nobuyoshi Suto, http://orcid.org/0000-0002-8994-2592
Amanda Laque, http://orcid.org/0000-0002-3650-0459
Eisuke Koya, http://orcid.org/0000-0002-5039-4875
Bruce T Hope, http://orcid.org/0000-0001-5804-7061
Friedbert Weiss, http://orcid.org/0000-0001-6211-6058

### Ethics

Animal experimentation: All experimental procedures were conducted in adherence to the National Institutes of Health Guide for the Care and Use of Laboratory Animals and were approved by the Institutional Animal Care and Use Committee of The Scripps Research Institute. (animal protocol #12-0032).

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
