## [Decision Letter]

Thank you for submitting your article "Distinct memory engrams in infralimbic cortex control opposing environmental actions on behavior" for consideration by *eLife*. Your article has been reviewed by two peer reviewers, including Steve Ramirez (Reviewer #1), and the evaluation has been overseen by a Reviewing Editor and Timothy Behrens as the Senior Editor.

The reviewers have discussed the reviews with one another and the Reviewing Editor has drafted this decision to help you prepare a revised submission.

This is an exciting paper that uses cutting-edge pharmacogenetic and behavioral techniques to assess the role of infralimbic (IL) neuronal populations in mediating opposing environmental actions (i.e. reward availability and omission) on behaviour (e.g. promotion and suppression of reward seeking). The authors bi-directionally modulate reward-seeking behaviour by identifying and inactivating IL neurons previously active either during reward availability of omission. They further show that cells in the IL increase their activity in response to both reward availability and omission cues, and they go on to successfully inactivate these ensembles to block cue-specific increases in neuronal activity. Overall, the experiments are well thought out, the paper is well written, and I view the current manuscript as an outstanding bridge between cells that differentially process discrete experiences and their causal contributions to reward seeking behaviour.

Essential revisions:

Please revise the paper to respond directly to the following reviewer comments and concerns:

1) If a control group in which a random set of cells of equal ensemble size (i.e. ~70 Fos-positive nuclei per mm^2^) is inactivated and the rats fail to show changes in lever-presses, then this result would greatly strengthen the claim that cells active during reward availability/omission per se are directly involved in modulating each behaviour. This experiment is important because an alternative interpretation of the data is that modulating any ensemble of equal size might thereby lead to behavioural changes in lever pressing without having to be directly tied to a defined c-Fos-inducing experience. I appreciate that this type of experiment has been performed previously (e.g. Koya et al. 2009) and, at the very least, warrants a brief discussion in the manuscript to convincingly argue that the behavioral perturbations observed are in fact tied directly to c-Fos-expressing cells during a defined epoch of time.

2) Are cells active during S+ or S- preferentially reactivated during S+ or S- presentations, respectively? This experiment would provide a positive control to demonstrate the "upper bounds" of S+ or S- induced reactivation of a defined set of IL cells and can be performed by first inducing β-gal expression in cells active in response to S+ or S-, followed by administering S+ or S- the following day and sacrificing the animals 1.5 hours later to measure β-gal and c-Fos overlap levels. These data would compliment the authors' data in in Figure 3 in which they demonstrate that they can block the S+ or S- induced increases in IL activity with Daun02-mediated inactivation of these cells.

---

## [Author Response]

*Essential revisions:*

*Please revise the paper to respond directly to the following reviewer comments and concerns:*

*1) If a control group in which a random set of cells of equal ensemble size (i.e. ~70 Fos-positive nuclei per mm^2^) is inactivated and the rats fail to show changes in lever-presses, then this result would greatly strengthen the claim that cells active during reward availability/omission per se are directly involved in modulating each behaviour. This experiment is important because an alternative interpretation of the data is that modulating any ensemble of equal size might thereby lead to behavioural changes in lever pressing without having to be directly tied to a defined c-Fos-inducing experience. I appreciate that this type of experiment has been performed previously (e.g. Koya et al. 2009) and, at the very least, warrants a brief discussion in the manuscript to convincingly argue that the behavioral perturbations observed are in fact tied directly to c-Fos-expressing cells during a defined epoch of time.*

This is an important question regarding whether the observed effects of Daun02 on the bidirectional modulation of reward seeking by S+ and S- are due to the “quality” (those specifically activated by either S+ or S-) or “quantity” (any population of ~70 Fos-positive nuclei per mm^2^) of IL neurons that were selectively disrupted by Daun02. We have now clarified the issues raised here in the main text. In short, we believe that the original experimental design and results already provided sufficient evidence to reject the alternative interpretation that “modulating any ensemble of equal size might thereby lead to behavioural changes in lever pressing without having to be directly tied to a defined c-Fos-inducing experience”.

More specifically, we believe that a sufficient internal control to address the quality vs. quantity question was already implemented in the original study. In the current study, each animal was trained to recognize three sets of cues predictive of rewards at varying probability (Figure 1): active lever and light-cue (50%), S+ (100%) and S- (0%). In different animals, Daun02 was then used to selectively disrupt neurons specifically activated by S+ or S-. In the group of animals that received Daun02 into IL following exposure to S+, Daun02 disruption of “S+ activated” neurons blocked the promotion of reward seeking (Figure 2) as well as neural activation (Figure 3) by S+, but spared the suppression of this behavior (Figure 2) as well as neural activation (Figure 3) by S-. In the group of animals that received Daun02 into IL following exposure to S-, Daun02 disruption of “S- activated” neurons blocked the suppression of reward seeking (Figure 2) as well as neural activation (Figure 3) by S-, but spared the promotion of this behavior (Figure 2) as well as neural activation (Figure 3) by S+. Importantly, Daun02 disruption of IL neurons activated by either S+ or S- did not significantly alter basal reward seeking due to active lever and light-cue (Figure 2).

Taken together, Daun02 disruption of IL neurons activated by either S+ or S- exclusively altered the behavioral response (Figure 2) as well as neural activation (Figure 3) uniquely linked to the targeted cue – thus establishing that the observed effects were due to the quality rather than quantity of neurons selectively disrupted. We thus believe that the current results sufficiently confirm that “the behavioral perturbations observed are in fact tied directly to c-Fos-expressing cells during a defined epoch of time (exposure to S+ or S-)”.

In addition, we would like to mention that the current results are in line with our previous findings that the quantity of cells silenced by Daun02 is not the critical factor behind Daun02’s effects on disruption of learned behaviors. For example, in rats with cocaine self-administration experience, exposure to novel contexts induces 3-fold more accumbens Fos than an exposure to a cocaine context. Yet, Daun02 disruption of neurons activated only by the cocaine context (but not novel context) disrupts context-induced reinstatement of cocaine-seeking (Cruz et al., J Neurosci 2014, 34: 7437–7446).

*2) Are cells active during S+ or S- preferentially reactivated during S+ or S- presentations, respectively? This experiment would provide a positive control to demonstrate the "upper bounds" of S+ or S- induced reactivation of a defined set of IL cells and can be performed by first inducing β-gal expression in cells active in response to S+ or S-, followed by administering S+ or S- the following day and sacrificing the animals 1.5 hours later to measure β-gal and c-Fos overlap levels. These data would compliment the authors' data in in Figure 3 in which they demonstrate that they can block the S+ or S- induced increases in IL activity with Daun02-mediated inactivation of these cells.*

Distinctively tagging S+ or S- activated neurons in the same *Fos-lacZ* rat requires the time-course of β-gal and cFos expression to be sufficiently distinct. In a pilot study leading up to the original publication of the Daun02/*Fos-lacZ* method (Koya et al. 2009, Nat Neurosci, 12: 1069-73), we found that this was not the case: the time-course of β-gal and cFos expression was very similar. Thus, unfortunately, the experiment suggested here is not technically possible in *Fos-lacZ* rats. Moreover, to our knowledge, no method currently exists to distinctly tag separate neuronal ensembles in rats. While such methods do exist for mice (e.g., TetTag mice, Reijimers et al., 2007, Science, 317:1230-3), replicating the current experiments in mice will be beyond the reasonable scope of the current study. However, we fully agree that this is an important issue that needs to be systemically addressed in a future study.

Nevertheless, based on the current results (Figure 2 and Figure 3), we speculate that a significant extent of the neuronal ensembles recruited by S+ and S- are mutually exclusive rather than overlapping. We thus believe that the answer to the question raised here (“Are cells active during S+ or S- preferentially reactivated during S+ or S- presentations, respectively?”) is “yes”. We have now clarified these issues in the main text.